# Multi-Scopic Cognitive Memory System for Continuous Gesture Learning

**DOI:** 10.3390/biomimetics8010088

**Published:** 2023-02-21

**Authors:** Wenbang Dou, Weihong Chin, Naoyuki Kubota

**Affiliations:** Graduate School of Systems Design, Tokyo Metropolitan University, Tokyo 193-0831, Japan

**Keywords:** continuous learning, gesture recognition, incremental learning, topological map

## Abstract

With the advancement of artificial intelligence technologies in recent years, research on intelligent robots has progressed. Robots are required to understand human intentions and communicate more smoothly with humans. Since gestures can have a variety of meanings, gesture recognition is one of the essential issues in communication between robots and humans. In addition, robots need to learn new gestures as humans grow. Moreover, individual gestures vary. Because catastrophic forgetting occurs in training new data in traditional gesture recognition approaches, it is necessary to preserve the prepared data and combine it with further data to train the model from scratch. We propose a Multi-scopic Cognitive Memory System (MCMS) that mimics the lifelong learning process of humans and can continuously learn new gestures without forgetting previously learned gestures. The proposed system comprises a two-layer structure consisting of an episode memory layer and a semantic memory layer, with a topological map as its backbone. The system is designed with reference to conventional continuous learning systems in three ways: (i) using a dynamic architecture without setting the network size, (ii) adding regularization terms to constrain learning, and (iii) generating data from the network itself and performing relearning. The episode memory layer clusters the data and learns their spatiotemporal representation. The semantic memory layer generates a topological map based on task-related inputs and stores them as longer-term episode representations in the robot’s memory. In addition, to alleviate catastrophic forgetting, the memory replay function can reinforce memories autonomously. The proposed system could mitigate catastrophic forgetting and perform continuous learning by using both machine learning benchmark datasets and real-world data compared to conventional methods.

## 1. Introduction

Two types of information are used in human communication: verbal information and non-verbal information. For a robot to understand human intentions, it is important it understands both types of information. Gesture recognition is widely used in many fields, including sign language recognition [1], human interaction [2], human emotion detection [3], and rehabilitation [4]. Zabihi et al. [5] proposed a Transformer for Hand Gesture Recognition (TraHGR) using surface electromyography to recognize hand gestures, and Fan et al. [6] proposed a method that combines a 3D Convolutional Neural Network (3DCNN) and a Multi-Scale-Decoder (MSD) to improve the gesture recognition accuracy without using multiple sensors. Yu Ding et al. [7] propose a dynamic hand gesture recognition method that uses fine-grained skeleton features for improved performance. The method uses a deep-learning-based approach to learn the features and classify the hand gestures automatically. The proposed method achieves high accuracy and robustness but requires a large amount of training data for optimal performance. C. Xiong et al. [8] propose an online gesture recognition method that uses streaming normalization to improve performance and adapt to changing lighting conditions. The proposed method uses a CNN to extract features and a Recurrent Neural Network (RNN) to classify the gestures. The method achieves high accuracy and adaptability but requires a continuous stream of data for optimal performance. Chan et al. [9] proposed a method that uses a convolutional neural network to recognize hand gestures in real time. They improved the recognition system so that it can recognize gestures within a few hundred milliseconds. However, the limitation of the proposed method is that it may have difficulty recognizing complex gestures with multiple fingers. Jia et al. [10] proposed a method that uses wearable inertial sensors and a CNN to recognize continuous hand gestures, and it can recognize gestures without being affected by external factors, such as lighting or background. However, it may have difficulty recognizing gestures that involve subtle finger movements. Current research in gesture recognition has achieved excellent recognition accuracy. Still, the current situation is that the system can recognize only the categories predefined in the training phase during recognition but cannot be extended by adding new categories after model training. In addition, for more accurate gesture recognition accuracy, it is necessary to keep all learned data in the system and combine it with a new class of data to retrain the model from scratch. This consumes a large amount of memory and significantly increases the computational cost each time the system is retrained. Therefore, to better recognize a user’s new gesture without retraining the recognition system, the recognition system must maintain the knowledge it has learned while training new gesture data.

Conventional deep learning algorithms based on Deep Neural Networks (DNNs) have demonstrated excellent performance in image processing and speech recognition and have been applied in various situations. However, conventional DNNs are not capable of continuously learning many different tasks. Therefore, continuous learning of new jobs is an important issue for existing models in applications to robots that interact with humans daily, where environmental conditions and human states are constantly changing. When conventional DNNs learn a new task, they learn new knowledge and forget previously learned knowledge at the same time because the network structure is fixed. This is a problem known as “catastrophic forgetting” in deep learning [11]. To overcome these problems, the system must have a flexible structure and be able to learn new knowledge while simultaneously enhancing the retention of current knowledge in response to continuous new input because humans can continuously learn new knowledge and accumulate experiences in changing situations. Thus, in order to better approximate human learning and cognitive mechanisms, research into continuous learning systems with flexibility has been frequently conducted in recent years. Shmelkov et al. [12] proposed “hard attention to the task,” a gating mechanism that focuses the model’s attention on the current task to mitigate the risk of forgetting. This method achieved state-of-the-art results on various benchmarks. Shin et al. [13] presented a computationally efficient approach to continual learning, utilizing a generative model to replay past experiences while learning new tasks without forgetting old ones. Li and Hoiem [14] proposed a selective preservation method for continual learning, which outperformed state-of-the-art methods on various benchmarks and can be applied to different network architectures. Nguyen et al. [15] presented a replay buffer-based method for avoiding catastrophic forgetting, which is computationally efficient and applicable to various models. Venkatraman et al. [16] proposed a Bayesian inference-based approach for continual learning, which models the posterior distribution of the model’s parameters and continuously updates them to handle non-stationary environments in real-world applications. The stability–plasticity dilemma [17,18], which has been widely studied in both biological systems and computational models, states that a system must be both plastic and stable to learn new information without forgetting existing knowledge. This allows us to divide the methods of continuous learning into three categories: dynamic architecture, regularization, and memory replay.

(1) Dynamic Architecture: This method utilizes a network architecture with a flexible structure that is not fixed and can change dynamically in response to new inputs. B.Fritzke et al. [19,20] proposed the Grow Neural Gas (GNG), which complicates the network structure from a simple structure for continuous input, and Parisi et al. [21] proposed a dual-memory network in which nodes are inserted into the network over time for continuous object recognition. Parisi et al. [22] also proposed a Growing When Required network (GWR) for continuous human action recognition. In the same way, Chin et al. [23] proposed an episodic memory network that can generate topological maps over time for robot navigation. (2) Regularization: Regularization constrains the objective function during learning by adding a regularization term to prevent catastrophic forgetting [24,25]. The updating of the weights of the network structure is constrained by the regularization component included in the objective function. (3) Memory replay: Memory replay is a method that usually uses generative models such as Variational Autoencoders (VAE) [13] and Generative Adversarial Networks (GAN) [26]. VAE is a probabilistic graphical model consisting of two parts: an encoder, which models a low-dimensional latent space representative of high-dimensional data, and a decoder, which inversely maps the encoder by learning to recover the original data from the latent space. The GAN consists of a generative network, G, and a discriminative network, D. Networks G and D play a MiniMax game to learn the entire network. During the learning phase, G generates pseudo-data from the input data, while D aims to distinguish between pseudo-data and original data.

In this research, we propose a Multi-Scopic Cognitive Memory System (MCMS), a hierarchical memory network for continuous learning, considering the three methods mentioned above. The proposed system mimics human statement memory and consists of two layers: an episodic memory layer and a semantic memory layer. The proposed system learns on regularized input data and recognizes multiple data classes in real-time. To consider the stability and flexibility of the system, each layer in the system uses a dynamic topological architecture that allows the network size to be dynamically increased or decreased according to the input data. Furthermore, to maintain the acquired knowledge, there is a memory replay mechanism that provides the weights of the nodes in the episodic memory layer as input to the entire system for memory enhancement when there is no external data input. These allow the proposed approach to alleviate the catastrophic forgetting problem of conventional deep learning algorithms when continuously learning new data.

## 2. Multi-Scopic Cognitive Memory System

The proposed Multi-Scopic Cognitive Memory System (MCMS) mimics human declarative memory with a two-layer structure consisting of an episodic memory layer and a semantic memory layer, allowing for continuous learning like humans. The episodic memory layer is composed of an Infinite Echo State Network [27,28], which encodes data features into nodes through unsupervised learning. The learning process of the proposed system involves clustering the spatiotemporal features of the input data using unsupervised learning in the episodic memory layer to generate a topological map, while simultaneously memorizing the activation patterns of the nodes as the spatiotemporal features of the data. In the semantic memory layer, supervised learning is performed using both bottom-up inputs from the episodic memory layer and top-down input from the teacher labels. If new data matches the previously learned knowledge in each layer, the system maintains its stability by updating the weights of the nodes. In contrast, if the new data does not match, new nodes are generated by the rules of each layer to update the network structure flexibly and continue learning. To optimize the network size, inactive nodes that are not activated for a long time are removed during the learning process. Furthermore, to reduce catastrophic forgetting, the proposed system automatically replays the memorized knowledge acquired using the activation patterns of previously learned nodes in the episodic memory layer to strengthen the overall memory of the system. Figure 1 shows the architecture of the proposed MCMS, and Table 1 shows the network notation.

### 2.1. Episodic Memory Layer

The episodic memory layer consists of an Infinite Echo State Network, which can dynamically change the size of the network in response to input data. New nodes are created in the network to represent new input data, and edges are created to connect existing nodes in the network. As an initialization, the episodic memory layer generates two recurrent nodes based on the input data of the network. It consists of a weight vector $\omega_{j}$ that can represent the input data at each node j. To train the network, Equations (1) and (2) are used to determine the best winner node that can best represent the input x(t) at time t.
(1)$$A_{j}\left(t\right)=\exp\left[-\frac{\|x\left(t\right)-\omega_{j}\|{}^{2}}{2\sigma_{j}^{2}}\right]\cdot \exp\left[\frac{A_{j}\left(t-1\right)-1}{\sigma^{2}}\right],$$
(2)$$b=\arg\min\left(A_{j}\left(t\right)\right).$$

The activation value of the selected best winner node b is then calculated by using Equation (3).
(3)$$a_{b}\left(t\right)=\exp\left(-A_{b}\right).$$

If the activation value $a_{b}\left(t\right)$ is less than the preset threshold $a_{T}$, a new node N with a weight $\omega_{N}$ based on the input data is added to the network using the following Equation (4):(4)$$\omega_{N}=0.5\;\cdot \left(x\left(t\right)+\omega_{b}\right).$$

To connect the best winner node b with the second winner node, a new edge is created between them; $a_{b}\left(t\right)$ is greater than $a_{T}$, the best winner node b can represent the input data x(t), and consequently, the best winner node b and its neighbor n are updated using Equation (5) based on the input data x(t).
(5)$$\omega_{j,new}=c_{j}\;\cdot \;re_{j}\;\left(x\left(t\right)-\omega_{j,old}\right),$$
where $c_{j}$ is a contributing factor of node j, and $re_{j}$ is a regularity counter of node j. If no edges exist between the best winner node b and the second winner nodes, new edges are generated to connect them. Each edge is assigned an age counter incremented by one for each study. The age counter of newly generated edges is initialized to 0. A node is removed from the network if its activation counter exceeds the threshold and no other nodes and edges are connected. Nodes with age counters exceeding the threshold are also removed from the network.

In addition, each node in the network contains a regularity counter $re_{j}\in \left[0,1\right]$, which indicates the firing frequency rises over time. The regularity counter of the newly generated node is initialized as $re_{j}=1$. For each learning, the regularity counters of the best and second winner nodes are reduced using the following Equation (6):(6)$$\Delta re_{j}=\tau_{j}\;\cdot \;\lambda \;\cdot \left(1-re_{j}\right)-\tau_{j}.$$

By using this, the regularity of the nodes that have fired over time according to the input data is stored in each node. As a result, the node’s regularity counter can be associated with the temporal sequence importance stored in the node.

Since each node contains an age counter, nodes with age counters greater than the threshold and not connected by edges to other nodes are deleted from the network. We implement a new form of node deletion [29] using the following Equation (7) to prevent the removal of nodes that are necessary at the start of learning:(7)υ=μ(H)+σ(H),
where H is a vector representation of the regularity counters of all nodes in the network, μ is the mean function, and σ is the standard deviation.

The newly created node can be connected to the network only if $b_{J}<\rho_{b}\;\mathrm{and}\;re_{J}<\rho_{r}$. The nodes are updated using Equation (5) when the regularity counters satisfy the thresholds.

For a sequence of inputs, episodes are composed in the episodic memory layer. The episode memory layer stores distinct episodes that are linked to each other. Temporal connectivity is established to learn the regularity counters of the nodes in the network. The temporal connectivity represents the time sequence of nodes activated throughout the learning phase. The temporal connectivity between two consecutively activated nodes is incremented by one for each learning. If the best winner node b is activated at time t and the other nodes are activated at time t−1, their temporal connectivity is reinforced as in Equation (8):(8)$$T_{\left(b\left(t\right),b\left(t-1\right)\right),new}=T_{\left(b\left(t\right),b\left(t-1\right)\right),old}+1.$$

As a result, for each node m, the next node g can be obtained from the encoded time-series data by selecting the maximum value of temporal connectivity in Equation (9).
(9)$$g=\arg\max T_{\left(m,n\right)},$$
where n is the neighbor node of node m. Thus, the time-series activation order of the nodes can be automatically restored without the need for input data.

### 2.2. Episodic Memory Replay

A sequence of time-series memories can be generated for memory replay using the temporal connectivity of the nodes stored in the episodic memory layer. When learning new data in the network, the temporal connectivity stored in the episodic memory layer can be used for memory replay. For example, suppose the input data activate the best winner node b. In that case, the node with the largest regularity counter T is selected as the next node to be activated. For each node j, the memory to be regenerated with length $K^{em}+1$ is calculated by Equations (10) and (11).
(10)$$u\left(i\right)=\arg\max T_{\left(j,u\left(i-1\right)\right)},$$
(11)$$U_{j}=\langle \omega_{u\left(0\right)}^{em},\omega_{u\left(1\right)}^{em},\cdots ,\omega_{u\left(K^{s}\right)}^{em}\rangle ,$$
where T(i,j) is the time-series concatenation matrix, and u(0)=j, and $\omega_{u\left(i\right)}^{em}$ is the weight of node u(0) in the episodic memory layer. The time-series concatenation of nodes stored in the network can automatically generate a sequence of memories for memory replay to reinforce the knowledge in the network without previously trained data.

### 2.3. Semantic Memory Layer

The semantic memory layer is hierarchically interconnected with the episodic memory layer. The semantic memory layer consists of an Infinite Echo State Network that receives bottom-up input from the episodic memory layer and top-down input from semantic labels that store semantic information in a larger time scale. The semantic memory layer performs supervised learning, which is different from the episodic memory layer, by using obtained semantic information associated with the top-down input of semantic labels.

The mechanism of the semantic memory layer is similar to the episodic memory layer but with additional constraints for creating new nodes in the network. Learning of nodes in the network occurs when the network correctly predicts the class of the input labeled from the episodic memory layer at training time. If the predicted class is incorrect, a new node is generated. This additional condition suppresses the update rate of the nodes in the semantic memory layer. Furthermore, because of the hierarchical learning of input data, the nodes in the semantic memory layer can store knowledge for longer than those in the episodic memory layer. The semantic memory layer determines the best winner node based on Equations (12) and (13), using the best winner node of the episodic memory layer as input.
(12)$$A_{j}^{sm}\left(t\right)=\exp\left[-\frac{\|x\left(t\right)-\omega_{j}^{sm}\|{}^{2}}{2\sigma_{i}^{2}}\right]\cdot \exp\left[\frac{A_{j}^{sm}\left(t-1\right)-1}{\sigma^{2}}\right],$$
(13)$$b^{sm}=\arg\min\left(A_{j}^{sm}\left(t\right)\right),$$
where x(t) is the input of the best winner node from the episodic memory layer, and x(t) is replaced by $\omega_{b}^{em}$ at the learning phase. The best winner node in which the semantic memory layer is selected is the node with higher similarity with the input from the episodic memory layer than the other nodes. A new node is created only if the best winner node b does not satisfy the following three conditions: $\left(1\right)\;a_{b}^{em}\left(t\right)<\rho_{a},\left(2\right)\;r_{b}^{sm}<\rho_{r},\left(3\right)$ the label $\zeta_{b}^{sm}$ of the best winner node is different from the label ζ of the data input. If the input from the episodic memory layer is not labeled, the semantic memory layer ignores matching with this label. Suppose the best winner node b predicts the same label $\zeta_{b}$ as the label ζ of the input. In that case, the node learning occurs by an additional learning factor ψ=0.001, and Equation (5) becomes the following Equation (14),
(14)$$\omega_{j,new}^{sm}=\psi \;\cdot \;c_{j}\;\cdot \;re_{j}\;\cdot \left(\omega_{b}^{em}-\omega_{j,old}^{sm}\right).$$

This allows the semantic memory layer to maintain network stability while decreasing the learning rate and allows learning over long periods by associating the episodic memory layer with semantic labels.

## 3. Experiments and Results

To demonstrate the effectiveness of the proposed MCMS in continuous learning, we trained and validated the system using the conventional machine learning benchmark datasets Banknote, Iris, Seed, and Wine, which can be downloaded from the UCI Machine Learning Repository [30]. To compare the learning results with conventional continuous learning algorithms, we compared the learning results with GWR (context = 2) proposed by Parisi et al. [22] which has a two-layer structure similar to the proposed system. In addition, continuous learning was performed on each dataset. Thus, in the comparison experiments, MCMS and GWR learn sequential input for each data class in each dataset, and in each dataset, GWR performs memory replay for each batch, while the proposed MCMS performs memory replay for each data class.

The results of the comparison experiment are shown in Table 2. The experimental results show that the proposed MCMS and GWR can continuously learn and recognize classes after learning. Among the four datasets, the proposed MCMS outperforms the conventional GWR in terms of accuracy, precision, recall, and F-measure for Banknote, Iris, and Seed. It was also observed that in the Wine dataset, GWR had a higher accuracy percentage than the proposed MCMS, but the accuracy of the proposed MCMS also reached 90%.

To further demonstrate the learning capability of the proposed MCMS, we collected a hand gesture dataset. We conducted continuous learning experiments comparing a three-layer deep learning DNN with a two-layer GWR with different contexts (context = 1,2,3). Hand gesture data were collected using RGB images from a webcam, and posture extraction was performed using the open-source framework Mediapipe [31]. The collected hand gesture dataset uses input data from Mediapipe with a skeleton coordinate as the feature vector. Specifically, as shown in Figure 2, hand skeleton coordinates are extracted from RGB images using Mediapipe. Since the acquired skeleton coordinates of the hand depend on its position in the image, they are transformed to a relative coordinate system with the origin at the wrist, using the skeleton coordinates extracted using Equation (15),
(15)$$\{\begin{array}{l} x\left[i\right]=x\left[i\right]-x\left[0\right]\\ \;y\left[i\right]=y\left[i\right]-y\left[0\right] \end{array}\left(i\;is\;the\;number\;of\;each\;joint\right).$$

As shown in Figure 3, we collected the following ten hand gestures dataset, using the Chinese counting gesture as an example. The dataset consists of 400 training data for each hand gesture, including left and right hand, and 100 test data. Table 3 also shows the parameters of the proposed MCMS set up for this experiment. During comparing experiments, the parameters of GWR are set as in the original work, and the size of each layer of DNN is set as (40,100,10).

Sequentially, we input mini-batch data (100 data) into the proposed MCMS, DNN, and GWR for continuous learning training, and these data would not be used again. Because of this training cycle, the learning epoch is set to one so that all networks only look at the data once. In addition, since the proposed MCMS and GWR have a memory replay function, memory replay will be performed during the learning phase. However, GWR performs memory replay for each mini-batch, while the proposed MCMS performs memory replay only for each data class. A validation experiment was conducted after learning each data class to compare each method, and the results are shown in Figure 4 and Figure 5. In the continuous learning algorithm GWR, the results for context are set to two and are superior to the others. Thus, we compare the proposed MCMS and GWR (context = 2) after all data class training, and the comparison results are shown in Table 4.

## 4. Discussion

The experimental results on the benchmark datasets show that the proposed MCMS outperforms the conventional continuous learning method GWR on most datasets. In addition, during the experiment, MCMS performs memory replay for each new data class, while the conventional GWR performs memory replay for each batch. This indicates that the presented time of data in MCMS is far less than GWR during the entire learning phase. However, we found that the input data of the proposed MCMS needs to be normalized, which may cause a drop in accuracy in learning high-dimensional data (Table 2 Wine data). As shown in Figure 4, the conventional deep learning algorithm suffered from catastrophic forgetting due to continuous learning in the hand gesture recognition experiment. The previous memory could not be preserved after learning a new class. The DNN’s accuracy decreased with the increase of data classes to be learned. On the other hand, MCMS showed a decrease in accuracy with an increase in the number of data classes, but it reduced the forgetting of the learned data classes. Furthermore, looking at the features of each gesture, the gesture from 1 to 5 shows that the learning of the features is reduced in the learning of the proposed system because the gesture from 1 to 5 shows few changes in the fingers. However, since the proposed system uses the memory replay function, it is seen that the recognition rate is improved because the learning of the previously learned gestures is enhanced because of the intense changes in the feature values of each gesture in learning new data from gesture five onward. In addition, the same learning procedure was applied to the hand gesture dataset, and the results showed that the proposed MCMS learned a higher recognition rate than the conventional GWR. The results of the comparison of each layer showed that the proposed MCMS outperformed the GWR in the recognition rate for 10 classes of data by comparing the episodic and semantic memory layer of the proposed MCMS and the GWR. At the same time, we observed that the proposed MCMS shows a decrease in accuracy as the class of training data increases, but the decrease is slighter than the conventional GWR. These results indicate that the proposed MCMS can reduce the catastrophic forgetting of deep learning in continuous data learning, and the performance of the proposed MCMS in continuous learning can be considered higher than that of the conventional continuous learning method GWR.

## 5. Conclusions

In this study, we proposed a Multi-Scopic Cognitive Memory System (MCMS) with two layers of episodic memory layer and a semantic memory layer for continuous learning gesture data. The proposed system was able to learn and recognize multiple class data inputs continuously. Experimental results on real data showed that the proposed MCMS could alleviate the conventional catastrophic forgetting problem, demonstrating the effectiveness of continuous learning. While this study performed continuous learning on hand gesture data, future work will require verification of continuous learning on data with spatiotemporal representations. For example, the system will learn gesture and action patterns in real-time by action segmentation of time-series data such as human behavior patterns. In addition, considering the practicality of the proposed MCMS system for robots, it is necessary to construct a continuous learning and recognition system that can deal with multiple types of data from different sensors by using multiple channels.

## Figures and Tables

**Figure 1 biomimetics-08-00088-f001:**
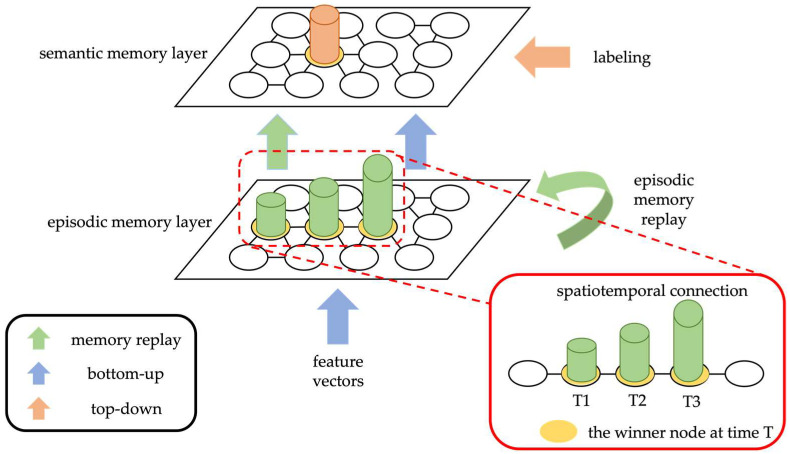
The architecture of the proposed MCMS system is as follows: the episodic memory layer learns the spatiotemporal features of the data through unsupervised learning. The semantic memory layer performs supervised learning using the output from the episodic memory layer and labels. Furthermore, to reinforce existing knowledge, the system automatically replays the memories using the nodes in the episodic memory layer to facilitate the overall relearning of the system.

**Figure 2 biomimetics-08-00088-f002:**
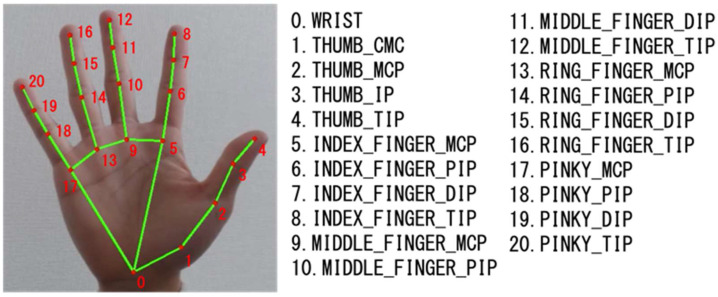
Example of skeleton coordinates by Mediapipe.

**Figure 3 biomimetics-08-00088-f003:**
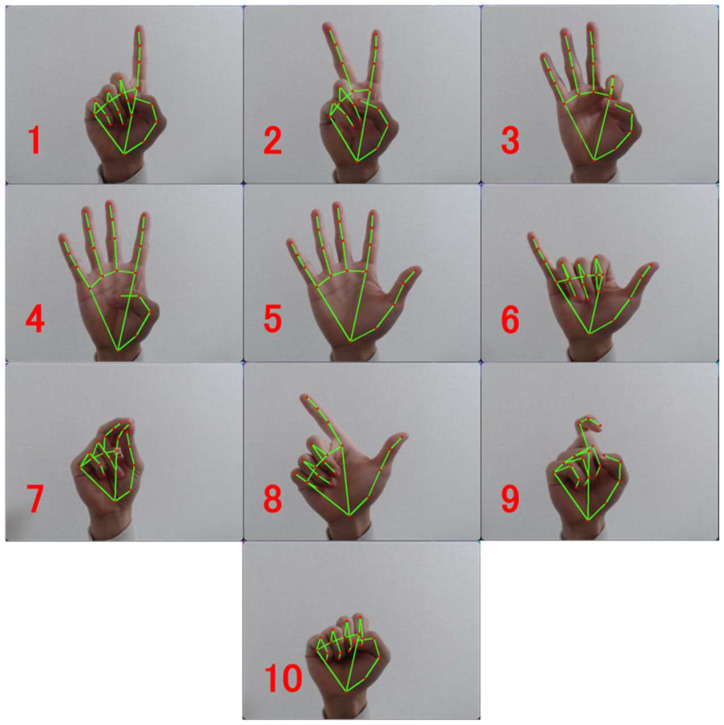
Example of a Chinese counting gesture. The meaning of each gesture is shown in red.

**Figure 4 biomimetics-08-00088-f004:**
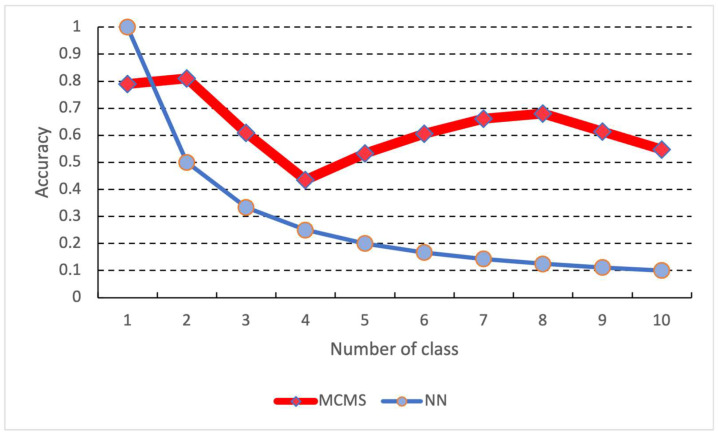
Comparison results between the proposed MCMS and DNN.

**Figure 5 biomimetics-08-00088-f005:**
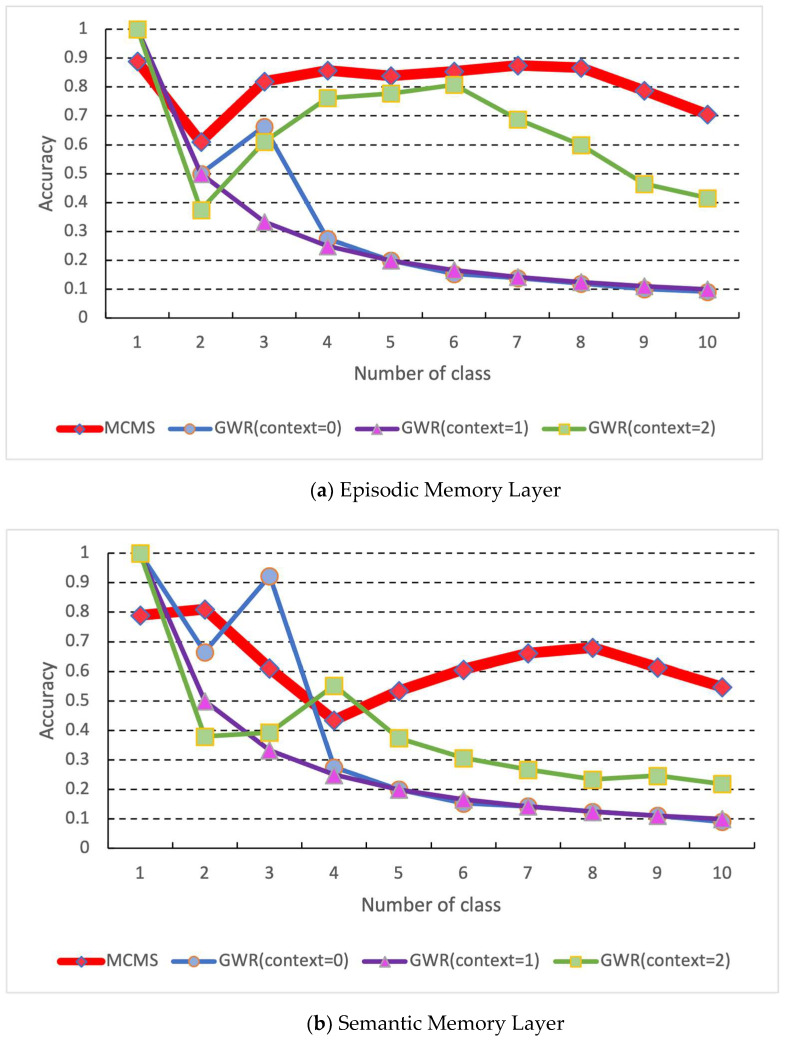
Comparison results of each layer of the proposed MCMS and GWR: (**a**) episodic memory layer; (**b**) semantic memory layer.

**Table 1 biomimetics-08-00088-t001:** Notation of the proposed MCMS.

Notation	Definition
$A_{j}\left(t\right)$	Activation value of node *j* at *t*
$\omega_{b}\left(t-1\right)$	Best matching node weights at *t* − 1
$re_{j}$	Regularity counter of node *j*
$c_{j}$	Contributing factor of node *j*
$\tau_{j},\lambda$	Decay factors for regularity counter
$\sigma_{i}^{2},\sigma^{2}$	Kernel width
ρ	Learning threshold
T(m,n)	Temporal connection between node *m* and *n*
L	Associative matrix for labeling
*b*	Index of best matching node

**Table 2 biomimetics-08-00088-t002:** Experimental results comparing the proposed MCMS and GWR.

Title 1	Title 2	Banknote	Iris	Seed	Wine
MCMS	accuracy	0.931	0.900	0.850	0.900
	precision	0.933	0.923	0.865	0.923
	recall	0.931	0.900	0.850	0.900
	F1	0.931	0.897	0.847	0.897
GWR(context = 2)	accuracy	0.671	0.666	0.666	0.966
	precision	0.790	0.500	0.718	0.969
	recall	0.671	0.666	0.666	0.966
	F1	0.619	0.555	0.642	0.966

**Table 3 biomimetics-08-00088-t003:** Parameter setting for the proposed MCMS.

Parameter	Value
$\rho_{a}$	0.85 (episodic), 0.80 (semantic)
$\rho_{r}$	0.1
$\tau_{j}$	0.5
λ	1.05
$\sigma_{i}^{2},\sigma^{2}$	1.5, 1.2
$c_{b}$	0.2
$c_{n}$	0.001
$re_{e}$	0.001

**Table 4 biomimetics-08-00088-t004:** Comparison results between the proposed MCMS and GWR (context = 2) by using all classes of data.

	GWR	MCMS
Accuracy	0.219	0.548
Precision	0.192	0.463
Recall	0.219	0.548
F1	0.130	0.473

## Data Availability

Publicly available datasets were analyzed in this study. This data can be found here: [https://archive.ics.uci.edu/ml/index.php, accessed on 26 January 2023]. The gesture data presented in this study are available on request from the corresponding author. The data are not publicly available due to privacy issues.

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
