# Peer review of "Multi-Scopic Cognitive Memory System for Continuous Gesture Learning"

_biomimetics, 2023, doi:10.3390/biomimetics8010088_

Round 1

Reviewer 1 Report

Dear Authors,

I think it is of an author's best interest to have a review with the highest amount of fair-criticism as possible, thus having his/her name associated with high-quality work. Minding the time constraints to review this paper, I spent the maximum amount of time I could on it and tried to be as critical as I could.

As a Reviewer my general opinion is that the topic is interesting.

However, I think the manuscript is not ready yet for publication, it still needs maturing in terms of the accuracy in theory and also in formatting.

Next, see my complete review, minor and major issues are blended. The comments follow the same order as the paper.

1) I think that your Abstract is well-written. But you should be more specific regarding the methods and the implementations which are tested and proposed in your work. As I can observe, you provide more general information in abstract, please be more specific. Are the methods new in the domain or new in the research? Thank you.

2) Please provide extra information related to previous works and design methodologies. You just refer to previous related works. Please extend your introduction and provide more state-of the art information. It will increase the quality of your work and you will receive easier citations. Moreover, add a Reminder of your work. Please also be careful with abreviations!

3)Please explain more Figure 1. Your explanation is a little bit weak. If I read only your text, it will not be easy to understand that you refer to this Figure. Moreover, please provide extra information in Figure 1 caption. As a result, please extend both the main text and the caption related to Figure 1.

4) Regarding the Multi-Scopic Cognitive Memory System, please explain more the necessity of this methodology. Please provide application example and real-world implementations. Also, the limitations of each methodology are not clearly provided.

5)Please check again all the mathematical equations. You should be 100% sure that a reader can use them and clearly understand the methodology.

6) Please provide information and comparison related to recent software and hardware based works (new related works). For example:

a) A structured and methodological review on vision-based hand gesture recognition system

b) Real-time hand gesture recognition using fine-tuned convolutional neural network

c) A Hand Gesture Recognition Circuit Utilizing an Analog Voting Classifier

d) An Integrated Real-Time Hand Gesture Recognition Framework for Human–Robot Interaction in Agriculture

7)Which are the values for the parameters (training)?

8) Have you checked your methodology over extreme cases and difficult datasets?

9) Fig. 5 please create new figures in which you combine each part. It will help the reader. Now the resolution is very weak.

10) Please provide a Table in which you compare your work with previous related works. It will upgrade your work. (related metrics). Please also explain Table 2. Have you checked all these datasets in the same algorithm?

11) Please provide the difficulties of the implemented architecture (for example tuning in training etc). Thank you.

12)     How you can use this design flow in a different application level (different types of datasets) ? Is it a simple model? Can you use it as a part of a whole general purpose system or just as a single application specific one? Please explain.

13)  Moderate English changes required. Please ask for help from a native American/English speaker, if it is possible.

Please carefully deal with my concerns, this will upgrade your work!

Author Response

We thank reviewer for the constructive comments. Please see the attachment.

Reviewer 2 Report

This paper studies and improves the gesture recognition accuracy of intelligent robots. The author proposed a Multi-scopic Cognitive Memory System (MCMS, a hierarchical memory network for continuous learning) to solve the problem of catastrophic forgetting that cannot be sustained by traditional deep learning algorithms. Each layer of the MCMS uses a dynamic topology that dynamically increases or decreases based on input data, enabling continuous learning like humans. According to the experimental results, it can be seen that MCMS greatly improves the accuracy of gesture recognition. However, there are also some problems in this article, as follows:

1.     The author should pay attention to the citation number of the paper reference. For example, the reference in line 219 is unlabeled.

2.     I suggest that Fig 4 and 5 be slightly retouched to make them more beautiful.

3.     It is suggested that the chart and table should be typeset again, and the table should not appear across pages as far as possible.

Author Response

(The authors gave the same response as above.)

Round 2

Reviewer 1 Report

Dear Authors,

Thank you for your response. I hope to see your work published soon.

Best Regards.